# Virtual Sensors for Optimal Integration of Human Activity Data

**DOI:** 10.3390/s19092017

**Published:** 2019-04-29

**Authors:** Antonio A. Aguileta, Ramon F. Brena, Oscar Mayora, Erik Molino-Minero-Re, Luis A. Trejo

**Affiliations:** 1Tecnologico de Monterrey, Av. Eugenio Garza Sada 2501 Sur, Monterrey, NL 64849, Mexico; 2Facultad de Matemáticas, Universidad Autónoma de Yucatán, Anillo Periférico Norte, Tablaje Cat. 13615, Colonia Chuburná Hidalgo Inn, Mérida, Yucatán 97110, Mexico; 3Fandazione Bruno Kessler Foundation, 38123 Trento, Italy; omayora@fbk.eu; 4Instituto de Investigaciones en Matemáticas Aplicadas y en Sistemas—Sede Mérida, Unidad Académica de Ciencias y Tecnología de la UNAM en Yucatán, Universidad Nacional Autónoma de México, Sierra Papacal, Yucatán 97302, Mexico; erik.molino@iimas.unam.mx; 5Tecnologico de Monterrey, School of Engineering and Sciences, Carretera al Lago de Guadalupe Km. 3.5, Atizapán de Zaragoza, Estado de México 52926, Mexico; ltrejo@tec.mx

**Keywords:** optimal data integration, virtual sensors, fusion methods

## Abstract

Sensors are becoming more and more ubiquitous as their price and availability continue to improve, and as they are the source of information for many important tasks. However, the use of sensors has to deal with noise and failures. The lack of reliability in the sensors has led to many forms of redundancy, but simple solutions are not always the best, and the precise way in which several sensors are combined has a big impact on the overall result. In this paper, we discuss how to deal with the combination of information coming from different sensors, acting thus as “virtual sensors”, in the context of human activity recognition, in a systematic way, aiming for optimality. To achieve this goal, we construct meta-datasets containing the “signatures” of individual datasets, and apply machine-learning methods in order to distinguish when each possible combination method could be actually the best. We present specific results based on experimentation, supporting our claims of optimality.

## 1. Introduction

The pervasive use of sensors is a natural consequence of both their usefulness and the simple fact that they are increasingly cheap and available. The proliferation of sensors and processors has been called “ubiquitous computing” or “pervasive computing” [1,2,3]. As useful as they are, though, sensors are very often subject to failures (such as sensor deprivation, limited spatial coverage, imprecision, and uncertainty [4]) or simply generate noisy signals, so that the readings given by sensors could not be taken at face value.

One way of dealing with the lack of reliability of sensors is to use redundancy, which in its simplest form could be just using several sensors of the same kind, for instance, in order to average the readings and obtain a better signal-to-noise ratio. However, it is also possible to use a combination of several sensors, which ideally would have complementary strengths, in order to obtain a better overall information about the phenomenon being observed.

We use the term “virtual sensor”, which have been introduced in restricted contexts elsewhere [5,6,7,8,9], to refer to any set of physical sensors from which we extract information that we call “features”, and then the features are combined to obtain the high-level information we are interested in. Virtual sensors are no longer physical devices, but abstract aggregations of the information coming from actual sensors. If constructed properly, virtual sensors could be much more precise and reliable than their physical counterparts, as we will verify in the following.

In particular, in the context of Human Activity Recognition (HAR; in this paper, we use the term “activity” to refer to purposeful sequences of atomic actions: the activity of walking is composed of actions such as lifting one foot, putting it forward, etc.), it is normal to use a combination of sensors, often portable ones. HAR has received much attention in recent years because it is an important element to understand the user’s context [10]. These activities could be “simple” activities, for instance, walking, biking, sitting, running, climbing stairs, or “complex” ones, such as shopping, attending a meeting, having lunch or commuting [11]. The user‘s context is the information that can be used to distinguish the situation of a user [12]. Understanding this context is the basis for context-aware applications [12] that can deliver proactive and personalized services [13] in areas such as medicine, Ambient-Assisted Living [14], sports, marketing [15], surveillance [16], just to mention some. The sensors commonly used include accelerometers, gyroscopes, and many others [17,18]; in this study, we only considered information coming from accelerometers and gyroscopes taken from datasets reported by other researchers, so the present work could be considered as a meta-study.

With the aim of addressing the issues that arise when using one sensor, and of improving the performance (measured mainly by accuracy, recall, sensitivity and specificity [19]) of the recognition, researchers have explored the ideas that the information from one sensor can complement the shortcoming of the others [20], and also the imprecision of a single sensor can often be compensated by other ones [21]. These ideas have led to a wide variety of methods for combining the information coming from similar or different sensors, and there are active research areas called “Sensor Fusion”, “Information Fusion” and similar ones. However, the proliferation of information integration methods makes it difficult to understand why one method is better than the others in specific conditions, so when researchers want to integrate information coming from several sources, they have nothing better than trial and error -or worse, just to try the methods they are familiar with. The research opportunity we detected is the need to systematically find the best way to combine data from a given set of portable sensors.

When comparing the different information integration methods, in this paper, we are focusing only on improving the precision (and related measures such as accuracy), but not dealing with resilience to failure. We are also focusing our attention to data integration for the task of human activity recognition, though, as we will see, there is nothing about our method itself which restricts the application to this area, and perhaps later on we will claim more generality in our results.

It has been proposed to group sensor information integration methods into three main categories: *data-level* fusion, *feature-level* fusion, and *decision-level* fusion [22], which are going to be explained in the next section. Not all the information combination methods address the same issues: some address aspects such as increasing the performance (accuracy, recovery and specificity) of the recognition of the activity [23,24,25], improving the tolerance against sensor faults and sensor failures [26], or reducing the energy consumption of the sensors while ensuring accuracy of recognition [27].

This work aims to systematically find the optimal integration method of information derived from sensors, for the human activity data of two given portable sensors (like accelerometers and gyroscopes). We considered a specific set of eight information integration strategies, starting with the simplest one of constructing a long feature vector by concatenating the features derived from each sensor (we call it “aggregation”); this is considered as a baseline for comparison purposes.

After verifying that, in some specific contexts, different integration methods are better than the others (for example, Multi-view stacking method [28] reached an accuracy of 0.92, while Voting method [29] reached an accuracy of 0.88, for a dataset derived from Opportunity dataset [30]), and so there is no such thing as “the best integration method”, we guessed that there should be something in the data itself making one integration method or another the best one in that context. In order to verify this hypothesis, we gather a rather large number (65) of HAR-related datasets, and characterized each dataset with a vector of some statistical features, which we call “dataset fingerprint”, and also manually tagged which was the best integration method for that specific dataset, based on experiments. With these fingerprints, we constructed a meta-dataset and trained a classifier intended to learn to predict the best sensor information integration method for a given dataset.

As our experiments show below, we are confident to say that the classifier learned with good precision to predict the best integration strategy for a given dataset (not contained in the training set, of course). We believe that this is an original breakthrough, which can lead to practical applications.

We envision the application of our results in the architecture shown in Figure 1, where we can see that the analysis of datasets, their fingerprint processing and the training of the classifier to find out the best information integration method are all done offline; once the integration method is determined, it can be implemented in the “virtual sensor” itself with electronics and commodity electronics, which could run online (We have not built the electronic implementation, as all this work is done offline, but we argue that there is no obstacle for building it).

The contribution of this work is to propose a method for determining the best integration of information coming from sensors, with a precision above 90%, using a machine-learning approach that uses the statistical “fingerprint” of a collection of datasets, involving an accelerometer and a gyroscope. We are also proposing a specific architecture of how this method can be applied to build optimal “virtual” sensors.

This document is organized as follows: Section 2 presents the background and state of the art. Section 3 shows the proposed method. Section 4 shows the experimental methodology. Section 4.5 presents the results and discussions. Finally, in Section 5, we draw conclusions and present future work.

## 2. Background and State of the Art

In this section, we present the background and the state of the art, we discuss the recognition of human activity, the workflow of recognition of the activity and the fusion of data from multiple sensors.

### 2.1. Human Activity Recognition

Research on HAR has grown considerably in recent years [31,32,33,34] for its usefulness in areas such as medical applications. For example, Bernal et al. [35] propose a framework to monitor and assist a user perform a multi-step medical procedure. Kerr et al. [36] present an approach to recognizing sedentary behavior. O’Loughlin et al. [37] put forward a method to analyze dietary practices. Silva et al. [38] propound an approach to study the effectiveness of egocentric video as a memory aid. Tacconi et al. [39] proposed an approach for supporting diagnosis of bipolar disorder. Rad et al. [40] put forth a framework to the automatic Stereotypical Motor Movements detection. HAR is also used to detect falls and other forms of help for elders [41,42,43,44]. Another application area of HAR is sports; for example, Wei et al. [45] propound a scheme on sports motion evaluation. Ahmadi et al. [46] present a method to assess all of an athlete’s activities in an outdoor training environment. In addition, Ghasemzadeh et al. [47] come up with a golf swing training system that provides feedback on the quality of movements. In addition, Ghasemzadeh et al. [48] put forward a framework that evaluates the swing of baseball players to provide a corrective feedback. In addition, we found other motivations to recognize human activities in the development of applications such as Ambient-Assisted Living [14], marketing [15], and surveillance [16].

### 2.2. Activity Recognition Workflow

To recognize activities in an orderly manner, professionals and researchers follow a sequence of similar steps during the implementation of the software for that purpose [49]. In Figure 2, we present this Activity Recognition workflow (ARW).

In the first step, the raw data are obtained by sensors, which can be external or portable [50]. Video cameras, microphones, motion sensors, depth cameras, RFID tags, and switches are example of external sensors. Accelerometers, gyroscopes, and magnetometers are instances of wearable sensors [51]. These raw data are sampled with the objective of generating multivariable time series. Some techniques to sample the raw data are fixed rate, variable rate, adaptive sampling, compressed sensing, and sensor bit-resolution tuning [52,53].

In the Processing step, the raw data of human activities are prepared for the extraction of features. This preparation must retain the raw data properties that are important in the discrimination of human activities [49]. In addition, this preparation must consider the sensor class, the raw data quality of the sensor and the corresponding software application [54,55]. Examples of algorithms commonly used to achieve this preparation are calibration, unit conversion, normalization, resampling, synchronization or signal level fusion [56].

In the Segmentation step, the processed data obtained from the previous step is divided into segments that contain information about human activities. Some commonly used segmentation methods are the sliding window [57], energy-based segmentation [58], rest-position segmentation [59], the use of one sensor modality to segment data of a sensor of another modality [60], and the use of external context sources [49].

The fourth step extracts the characteristics of the segmented data from the previous level and organizes them into vectors that together form the space of the characteristics. The idea behind these features is to increase the possibility of grouping characteristics that share the same activity in the feature space and separate the properties of the actions that are different. In addition, these characteristics should increase the possibility of grouping the attributes corresponding to the same activity, but of different subjects, or actions of the same person, but which performs in different ways [49]. Examples of these characteristics are the mean, variance, or kurtosis (statistical features). Other examples of features are the mel-frequency cepstral coefficients or energy in specific frequency bands (Frequency-domain features) [61]. In addition, we have features extracted from a 3D skeleton generated by body sensors (Body model features) [62]. Other instances of characteristics are encoded duration, frequency, and co-occurrences of data (expressive feature) [11,63].

In addition, in this step, the task of selecting features is performed because the reduction of them is essential to diminish computational resources. Because this task of manually choosing such features is complicated, several techniques have been developed to automate this selection [49], such as these can be categorized into wrapper [64], filter [65], or hybrid [66] methods.

On the other hand, it is important to bear in mind that, recently, deep models have been used to automatically generate and select features, such as Convolutional Neural Networks (CNNs) [67].

In the Training step, the inference algorithms are trained with the features extracted in the fourth step and the truth labels. During training, the parameters of these algorithms are learned by reducing the classification error [49]. Examples of commonly used inference methods are the k-NN (k-Nearest-Neighbor) [68], Support Vector Machines (SVM) [69], Hidden Markov Models (HMM) [70], Decision Tree Classifiers (CART) [71], Logistic Regression (LR) [72], Random Forest Classifier (RFC) [73], the Naive Bayesian approach (NB) [74], and Artificial Neural Network (ANN) [75], such as Convolutional Neuronal Networks (CNNs), and Long Short-Term Memory networks (LSTMs).

In the classification step, the model trained in the previous step is used to predict activities (mapping feature vectors with class labels) with a given score. The final classification can be done in many ways, such as choosing the highest score and letting the application choose how to use the scores [49].

### 2.3. Multi-Sensor Data Fusion on HAR

Multisensor fusion had its origins in the 1970s in the United States Navy as a technique to overcome military issues, such as improving the accuracy of motion detection of the Soviet Navy [76]. Nowadays, various applications use this technique, such as the supervision of complex machinery, medical diagnostics, robotics, video and image processing, and intelligent buildings [77].

This multisensor fusion technique, which is used in some analysis tasks, refers to the combination of the features extracted from data of different modalities or the decisions generated from these characteristics by classification algorithms [78]. The objective of this technique is to accomplish better accuracies and better inferences than a single sensor can achieve [79]. This method has several advantages, such as that it can enhance the signal-to-noise ratio, diminish ambiguity and uncertainty, increase robustness and reliability, enhance resolution, accuracy and discrimination hypotheses [80].

Regarding the level of abstraction of data processing, multi-sensor fusion is typically divided in three main categories: data-level fusion, feature-level fusion, and decision-level fusion [22]. These categories are defined as follows:

**Data-level fusion:** At the lowest level of abstraction, it is generally assumed that the base components, such as the communication, storage, and processing systems are reliable and that the aim of the fusion algorithms is to combine multiple homogeneous sources of raw data in order to achieve more precise, informative and synthetic fused data than the sources [81]. Studies on data-level fusion are mainly concerned with the design and implementation of noise elimination, feature extraction, data classification and data compression [82]. Some examples of data-level fusion are: raw data aggregation [40], and time-lagged similarity features [83].

**Feature-level fusion:** Feature sets extracted from multiple data sources (generated from different sensor nodes or by a single node equipped with multiple physical sensors) can be fused to create a new high-dimensional feature vector representing the input for the recognition step of classification/patterns [84]. In addition, at this level of fusion, machine learning, and pattern recognition, depending on the type of application, will be applied to vectors with multidimensional characteristics that can then be combined to form vectors of joint characteristics from where the classification is carried out [85]. Examples of methods that fit this category are: feature aggregation [21,23,24,25,86,87,88], Temporal Fusion [35], and SVM-based multisensor fusion algorithm [89].

**Decision-level fusion:** The decision-level fusion is the process of selecting (or generating) a hypothesis from the set of hypotheses generated by individual decisions (local, and often weaker) of multiple sensors [90]. Examples of methods that fit the fusion category at the decision level are: Adaboost [91], Voting [29], Multi-view Staking [28], Hierarchical Weighted Classifier [26], and Genetic Algorithm-based Classifier Ensemble Optimization Method [92].

These levels of fusion take its place in the ARW and, in doing so, they configure an extended version of it (see Figure 3). In Figure 3, the merging at the data level occupies the second position because the raw data of several sensors feed this level. The fusion at the feature level is located between the step of extraction and selection of the characteristics, and the training step since this training requires the features extracted from the sensors. The decision-level merger occurs both in the training stage and in the classification stage because the decisions of some classifiers are combined to make a final decision. Each of these levels of fusion can occur one at a time or a combination of them.

## 3. Method

Our method to predict the best method to fuse the activity data from a given set of wearable sensors is depicted in Figure 4. It consists of three parts: (1) a *comparison* stage in which we find the best fusion method for each of the 65 original datasets, (2) a *fingerprint dataset* construction that creates a meta-dataset with what we call the *digital signature* of the sensor data, tagged with the best fusion method obtained in the previous stage, and (3) a *machine learning prediction* stage in which we train, then use a classifier to infer the best fusion method. Next, we explain each part.

### 3.1. Finding the Best Fusion Method Using the Friedman Test and Holm Test

In this section, we present the classifiers, the fusion strategies and its configurations, and the main steps that we follow to get the best fusion method.

#### 3.1.1. Classifiers and Fusion Strategies

We used the Voting [29], Ensemble [93], Adaboost [91] algorithms and Multi-view stacking technique [28] as ensemble methods. We chose these ensemble methods because HAR researchers use them commonly and have shown good performance, in terms of accuracy, in the HAR literature. In addition, we used classifiers that fit the data linearly or nonlinearly [94] because we are interested in exploiting and combining the strengths of these two types of data adjustments. The linear classifier is Logistic Regression (LR) [72]. The nonlinear classifiers are Decision Tree Classifier (CART) [71], and Random Forest Classifier (RFC) [73]. We chose these classifiers as a representative sample of both types of data adjustments and because they have demonstrated their effectiveness, in terms of accuracy, in the HAR literature.

Moreover, we used three fusion strategies: at the level of features, level of decisions, and the combination of the previous ones (mixed fusion). The feature-level fusion was done by aggregating the features extracted from the signals from the sensors. The decision-level fusion was made by voting on the predictions of the classifiers, each of which was trained with a third of the combined characteristics (random or non-random) of the sensor data. The mixed fusion was carried out in two steps. First, the features of the sensor data are concatenated, and then decisions were made from votes of the predictions of these classifiers. The fusion at data level was beyond the scope of this work because we are interested in obtaining high-level information (features) from raw data from wearable sensors to create “virtual sensors”. The level-data fusion occurs at the raw data level of the sensors.

#### 3.1.2. Configurations of the Fusion Strategies

In this section, we describe the different configurations that served as the basis for comparing fusion strategies defined in Section 3.1.1.

**Aggregation of features:** We combined by column wise the features extracted of the sensor data (feature-level fusion). Then, we used RFC to predict the activities recorded by the sensors.

**Vote with shuffled features:** We took the features extracted from the sensors, combined them by columns, shuffled them, and divided them into three parts, each part for each instance of a classifier. Then, we used three RFC instances as the estimators and the Ensemble algorithm as the classifier (for a decision-level fusion). Finally, we inferred the activities recorded by the sensors.

**Vote:** We took the features extracted from the sensors, combined them by columns, and divided them into three parts, each part for each instance of a classifier. Then, we used three RFC instances as the estimator and the Ensemble algorithm as the classifier (for a decision-level fusion). Finally, we inferred the activities recorded by the sensors.

**Voting with RCF, CART and LR for all features:** We took the features extracted from the sensors and combined them by columns (feature-level fusion). The combined features fed each classifiers. Then, we defined RFC, CART and LR as the estimators and Voting algorithm as the meta classifier (for a decision-level fusion). Finally, we inferred the activities recorded by the sensors.

**Multi-View Stacking with shuffled features:** We took the features extracted from the sensors, combined them by columns, shuffled them, and divided them in three parts. Then, we defined three RFC instances as the base classifiers. Each instance of the base classifier was trained with some of these three parts and the predictions of these instances were combined by column-wise (features-level fusion). After that, we define RFC as meta classifier (for a decision-level fusion). Finally, we inferred the activities recorded by the sensors.

**Multi-View Stacking:** We took the features extracted from the sensors, combined them by columns, and divided them in three parts. Then, we defined three RFC instances as the base classifiers. Each instance of the base classifier was trained with some of these three parts and the predictions of these instances were combined by column-wise (features-level fusion). After that, we define RFC as meta classifier (for a decision-level fusion). Finally, we inferred the activities recorded by the sensors.

**Multi-View Stacking with RCF, CART and LR for all features:** We took the features extracted from the sensors and combined them by columns (features-level fusion). Then, we defined three classifier (RFC, CART or LR) as the base classifiers. Each base classifier was trained with these combined features and the predictions of these classifiers were combined by column-wise (features-level fusion). After that, we define RFC as meta classifier (for a decision-level fusion). Finally, we inferred the activities recorded by the sensors.

**Adaboost:** We took the features extracted from the sensors and combined them by column-wise (feature-level fusion). Then, we defined RFC as the estimator. After that, we selected the Adaboost algorithm as the classifier (for a decision-level fusion). Finally, we inferred the activities recorded by the sensors.

We wrote Python code in the Jupyter Notebook application [95] to create the functions that implements the fusion strategy configurations.

#### 3.1.3. Steps to Get the Best Fusion Method

Here, we process the raw data from the wearable sensors (such as the accelerometer and gyroscope) to extract its features. Then, these features are fed into diverse fusion methods: Feature Aggregation, Voting (with three configurations), Multi-view stacking (with three configurations), and AdaBoost. The performance (in accuracy terms) produced by each method is compared to the yield produced by the Aggregation method, which is considered a baseline for comparison purposes.

We use Friedman Rank test [96] and Holm’s post hoc test [97] to perform this comparison. Friedman Rank test verifies the hypothesis that in a set of k≥2 groups of dependent samples (in this case, these configurations of fusion strategies), at least two of the groups represent populations with different median values. Holm’s post hoc test tests the hypothesis that the ranking of the control method (in this case, the Aggregation configuration) is different from each of the other methods (in this case: Voting with Shuffled features, Voting, Voting with RFC-CART-LR for all features, Multi-view stacking with shuffled features, Multi-view stacking, Multi-view stacking with RFC-CART-LR for all features and Adaboost). According to Demsar [98], these non-parametric tests are safe and robust when statistically comparing diverse classifiers. The result of Demsar are valid in our comparisons because the internal processes of the fusion methods use classifiers. Finally, we manually get the best fusion method for the wearable sensors based on such comparisons.

### 3.2. Fingerprint Dataset

In this stage, we extract a digital signature of the data features of each one of the original 65 datasets we study in this work. This could be considered as a “meta-dataset” where each row corresponds to one of the 65 datasets, describing its statistical properties, with the idea that these properties, used in this dataset as features, could be enough for training a machine-learning classifier, and in this way predicting the best information integration method.

The fingerprint dataset includes as features the mean, the standard deviation, the maximum value and the minimum value; likewise, the 25th, 50th and 75th percentiles. Then, we label this digital signature with the best corresponding fusion method obtained in the previous stages (see Section 3.1).

### 3.3. Prediction of the Best Fusion Method

In this last part, we train RFC with the Fingerprint data set with the intention of learning to recognize the best method, from those considered here, to integrate the sensor information for a given data set. The details of how we do the training and the prediction are going to be described in the next section.

## 4. Experimental Methodology

As explained in the introduction, we intend to predict, based on the features of a given dataset, which is the best way of combining the features in order to get the best measures of precision. For this, in the following, we first make a comparative analysis of the different information integration methods, trying different configurations, classifiers, numbers of features, as well as other variations. Then, we proceeded to follow a machine-learning data driven method for actually predicting, among a set of information combination options, which is the best one for a given (not included in the training stage) dataset. We explain in the following each of these elements.

### 4.1. Datasets

We obtained 65 datasets from six benchmark datasets commonly used by the HAR researchers. These six datasets were chosen because they have records of human activities collected by diverse sensors, such as accelerometers and gyroscopes, located in different parts of the body of a subject. The procedure we followed to obtain these 65 datasets consisted of forming different pairs of an accelerometer and a gyroscope, with all the accelerometers and gyroscopes available in each of the benchmark datasets. These couples were differentiated by one of their elements and not by their order. In the following, we briefly describe each of the six benchmark datasets.

UDT dataset [99] was collected using a Kinect camera (Microsoft, Redmond, WA, USA) and a wearable inertial sensor with a three-axis accelerometer and a three-axis gyroscope. This dataset has 27 actions performed by eight subjects with four repetitions per action. The actions include: 1—swipe left, 2—swipe right, 3—wave, 4—clap, 5—throw, 6—arm cross, 7—basketball shoot, 8—draw x, 9—draw circle CW, 10—draw circle CCW, 11—draw triangle, 12—bowling, 13— boxing, 14—baseball swing, 15—tennis swing, 16—arm curl, 17—tennis serve, 18—push, 19—knock, 20—catch, 21—pickup throw, 22—jog, 23—walk, 24—sit 2 stand, 25—stand 2 sit, 26—lunge, and 27—squat. In this experiment, we considered the accelerometer and gyroscope data corresponding to the 27 actions performed by eight subjects with four repetitions per action. With the combining of those two sensors, we create one new dataset.The Opportunity dataset consists of daily activities recorded with multi modal sensors. The available database [30] contains recordings captured by 4 subjects. We considered the four locomotion activities included in the database: 1—stand, 2—walk, 3—sit, and 4—lie. The total number of instances is 2477. For our experiment, we used the data of the accelerometer and gyroscope of each of the five Inertial Measurement Unit (IMU) located in different parts of the body of the subjects: Back (Ba), Right Lower Arm (Rl), Right Upper Arm (Ru), Left Upper Arm (Lu) and Left lower Arm (Ll). With the combining of pairs of these sensors, we create 25 new datasets.The PAMAP2 dataset [100] was collected using three Colibri wireless IMUs (inertial measurement units, Trivisio, GmbH, Kaiserslautern, Germany): one IMU over the wrist on the dominant arm (Ha), one IMU on the chest (Ch) and one IMU on the dominant side’s ankle (An). In addition, this dataset included data from Heart Rate monitor: BM-CS5SR from BM innovations GmbH, Hörgertshausen, Germany. This dataset has 18 actions performed by nine subjects. The actions include: 1—lying, 2—sitting, 3—standing, 4—walking, 5—running, 6—cycling, 7—Nordic walking, 8—watching TV, 9—computer work, 10—car driving, 11—ascending stairs, 12—descending stairs, 13—vacuum cleaning, 14—ironing, 15—folding laundry, 16—house cleaning, 17—playing soccer, and 18—rope jumping. In this experiment, we considered the accelerometer and gyroscope data corresponding to these three IMUs for eight actions (1, 2, 3, 4, 6, 7, 16, 17) performed by nine subjects. With the combining of pairs of these sensors, we create nine new datasets.The MHealth dataset [101] contains body motion and vital signs recordings for ten volunteers of various profiles while performing 12 physical activities. These activities are 1—standing still, 2—sitting and relaxing, 3—lying down, 4—walking, 5—climbing stairs, 6—waist bends forward, 7—frontal elevation of arms, 8—knees bending (crouching), 9—cycling, 10—jogging, 11—running, 12—jump front and back. The raw data of these activities were collected by the Shimmer2 (Shimmer DCU Alpha, Dublin, Ireland) [102] wearable sensors. The sensors were placed respectively on the chest (Ch), the right lower arm (Ra) and the left ankle (La) of the subject and were fastened with elastic straps. For our experiment, we considered the acceleration and gyro data from the Ra sensor and the La sensor for the first eleven activities. With the data of these sensors, we create four new datasets.The DailySport dataset [103] includes motion sensor data of 19 daily and sports activities, each performed by eight subjects in their own way for 5 minutes. The sensors used were five Xsens MTx units (Xsens Technologies B.V., Enschede, Netherlands) placed on the Torso (To), Right Arm (Ra), Left Arm (La), Right leg (Rl) and Left leg (Ll). The daily and sports activities are 1—sitting, 2—standing, 3—lying on back, 4—lying on right side, 5—ascending stairs, 6—descending stairs, 7—standing in an elevator still, 8—moving around in an elevator, 9—walking in a parking lot, 10—walking on a treadmill with a speed of 4 km/h in a flat position, 11—walking on a treadmill with a speed of 4 km/h in a 15 deg inclined position, 12—running on a treadmill with a speed of 8 km/h, 13—exercising on a stepper, 14—exercising on a cross trainer, 15—cycling on an exercise bike in a horizontal position, 16—cycling on an exercise bike in a vertical position, 17—rowing, 18—jumping, and 19—playing basketball. In this experiment, we considered the accelerometer and gyroscope data corresponding to the five Xsens MTx unit for 19 daily and sports actions performed by eight subjects. With the data of these sensors, we create 25 new datasets.The HAPT dataset [104] includes motion sensor data of 12 daily activities, each performed by 30 subjects wearing a smartphone (Samsung Galaxy S II, Seoul, South Korea) on the waist during the experiment execution. The daily activities are 1—walking, 2—walking upstairs, 3—walking downstairs, 4—sitting, 5—standing, 6—laying, 7—stand to sit, 8—sit to stand, 9—sit to lie, 10—lie to sit, 11—stand to lie, 12—lie to stand. In this experiment, we considered the accelerometer and gyroscope to create one new dataset.

### 4.2. Feature Extraction

The original signals of the accelerometer and the gyroscope were segmented into windows of fixed length of 3 s each, without superposition, because this is the typical value for the activity recognition systems [105]. Then, characteristic measures (features) were computed for each window segment.

In this work, we extracted 16 characteristics for each sensor (accelerometer and gyroscope) that are: the mean value of each of the three axes, the standard deviation of each of the three axes, the max value of each of the three axes, the correlation between each pair of axes, the mean magnitude, the standard deviation of the magnitude, the magnitude area under the curve (AUC, Equation (Equation 1)), and magnitude mean differences between consecutive readings (Equation (Equation 2)). The magnitude of the signal represents the overall contribution of acceleration of the three axes (Equation (Equation 3)). These types of features were chosen because they have shown to produce good results for activity recognition tasks [32,63]:(1)$$\mathrm{AUC}=\sum_{t=1}^{T}magnitude\left(t\right),$$
(2)$$\mathrm{meandif}=\frac{1}{T-1}\sum_{t=2}^{T}magnitude\left(t\right)-magnitude\left(t-1\right),$$
(3)$$\mathrm{Magnitude}\left(x,y,z,t\right)=\sqrt{a_{x}{\left(t\right)}^{2}+a_{y}{\left(t\right)}^{2}+a_{z}{\left(t\right)}^{2}},$$
where $a_{x}{\left(t\right)}^{2}$, $a_{y}{\left(t\right)}^{2}$ and $a_{z}{\left(t\right)}^{2}$ are the squared accelerations at time interval *t* and *T* is the last time interval.

### 4.3. Comparison Procedure of the Fusion Strategies’ Configurations

In this section, we present the procedure that we follow to compare the configurations of fusion strategies (see the Section 3.1.2).

For each dataset (see Section 4.1), we obtained 24 accuracy samples for each configuration of the fusion strategies. We got these samples by executing 24 times each of the functions that implement these configurations of fusion strategies. These functions used the cross-validation technique with three folds. In the case of settings that shuffle features, these functions shuffle these characteristics according to the number of them of the dataset that is processing.We repeat the step 1 for each classifier (see Section 3.1.1).We build boxplots with the accuracies obtained in step 2 to observe if there are differences between some pair of these configurations.We perform the Friedman test [96] with the data obtained in step 2 to know if there are significant differences between some pair of these configurations.We perform the Holm test [97] with the data obtained in step 4 to know if there is a significant difference between the Aggregation and some other configuration. We summarize these differences in Table 1 and Table 2.We perform the Holm tests [97] with the data obtained in step 4 to know if there is a significant difference between the configurations that shuffle features and the settings that do not shuffle them. We summarize these differences in Table 3.We perform the Holm tests [97] with the data obtained in step 4 to know if there is a significant difference between the configurations that include Multi-view stacking and the settings that include voting. We summarize these differences in Table 4.

### 4.4. Recognition of the Best Configuration of the Fusion Strategies

In this section, we present the main steps of an experiment to recognize the best configuration of fusion strategies for each dataset considered in this work. These main steps we follow to recognize the best configuration of the fusion strategies for each dataset considered in this document are:We built a Fingerprint dataset. We define this Fingerprint Dataset as a labeled dataset that contains the properties that describe the datasets. These properties, which were extracted for each of the features (see Section 4.2) of the datasets (see Section 4.1), are the mean, the standard deviation, the maximum value, and the minimum value; likewise the 25th, 50th, and 75th percentiles. The labels (MultiviewStacking, MultiViewStackingNotShuffle, and Adaboost) of this Fingerprint Dataset were obtained from the results in Table 2. We chose the results of this table because they show the best configurations of the fusion strategies, for each dataset (see Section 4.1). They are the best configurations, since they present the greatest significant differences with respect to Aggregation configuration. In cases where there were no significant differences between these configurations and the Aggregation configuration, we took the latter as the best option. Thus, for these cases, we label with the string of Aggregation.We balance the Fingerprint dataset because this dataset presents a situation of imbalance of the classes (the number of observations in each class is different). This situation would result in a classifier issuing results with a bias towards the majority class. While there are different methods to address the problem of the imbalance of the classes [106], we chose the Upsampling approach. Upsampling raises the number of samples of the minority class through the use of multiple instances of the samples of this minority class. In particular, we used an upsampling implementation for python: the resampling module from Scikit-Learn [107]. This module was configured to resample the minority class with replacement, so that the number of samples for this class matches those of the majority class.Based on this Fingerprint dataset, we recognize the best configuration of fusion strategies using the RFC classifiers. This classifier was trained and validated with a three-fold cross-validation strategy. The performance of this classifier was measured in terms of accuracy, precision, recall, and f1-score.

### 4.5. Experimental Results

In Table 1, we summarize our observations about the significant differences between the Aggregation configuration and the rest of the configurations, for each of the databases considered in this experiment. We mark with a check mark a configuration that presents a significant difference, and that is to the right of the Aggregation configuration. In the same way, we mark with an *X* mark the configuration that presents a significant difference, and that is to the left of the Aggregation configuration.

In Table 1, taking into account the check mark, we can see that the multi-view stacking configuration with the shuffled features shows a significant difference compared to the aggregation configuration for 33 of 65 datasets. This observation suggests that these 33 datasets have some properties that favor this configuration that use RFC as a classifier. In addition, we can observe that the multi-view stacking configuration presents a significant difference compared to the aggregation configuration for 16 of 65 datasets. This observation suggests that these 16 datasets may have some properties that favor this configuration that use RFC as a classifier.

In addition, in Table 1, taking into account the *X* mark, we can see that the Voting configuration with the shuffled features, the Voting configuration and the Voting configuration with all features show significant differences compared to the Aggregation configuration, for the most datasets. This observation insinuates that these configurations do not benefit from the characteristics of most datasets here considered. Furthermore, we can observe that Multi-view stacking with RFC-CART-LR (Random Forest Classifier, Decision Tree Classifiers, and Logistic Regression) for all features presents a significant difference compared to the Aggregation configuration, for 22 of 65 datasets. This observation suggests that this configuration does not benefit from the characteristics of these 22 datasets.

In Table 2, we summarize our observations about the greatest significant differences between the configuration of the Aggregation and the rest of the configurations, for each of the databases considered in this experiment. These last configurations are to the right of the Aggregation and achieved the highest accuracies. We mark with a check mark these configurations. In Table 2, we can see that the multi-view stacking configuration with the shuffled features presents a significant difference compared to the Aggregation configuration, for 23 of 65 datasets. This observation suggests that these 23 datasets have some properties that favor this Multi-view stacking configuration that shuffles features. We can also observe that the Multi-view stacking configuration presents a significant difference compared to the Aggregation configuration, for 11 of 65 datasets. This observation suggests that these 11 datasets may have some properties that favor this Multi-view stacking configuration. In addition, we can see that the Adaboost configuration presents a significant difference compared to the Aggregation configuration, for the OpportunityBaAccLuGy dataset. This observation suggests that these datasets may have some properties that favor this Adaboost configuration which uses RFC as classifiers.

In Table 3, we summarize our observations about the significant difference between the the Multi-view stacking configuration that shuffles features and the Multi-view stacking configuration that does not shuffle features, for each dataset in which at least one of these Multi-view stacking configurations presented a significant difference with respect to Aggregation (see Table 1). We mark with a check mark if the Multi-view stacking configuration that shuffles features presented a significant difference compared to the Multi-view stacking configuration that does not shuffle features, and if this significant difference was in favor of the Multi-view stacking configuration that shuffles features. In the same way, we mark with an *X* mark if the Multi-view stacking configuration that shuffles features presented a significant difference compared to the Multi-view stacking configuration that does not shuffle features, and if this significant difference favored the latter.

In Table 3, we can see that, in 10 of 34 datasets, there are significant differences in favor of the Multi-view stacking configuration that shuffles features. We also note that there are not significant differences in favor of the Multi-view stacking configuration that does not shuffle features. These observations suggest that the Multi-view stacking configuration that shuffles the features could be better than the multi-view stacking configuration that does not shuffle the features, depending on the way to shuffle the features of the different sensors.

In Table 4, we summarize our observations about the significant difference between the the best of the configurations that include Multi-view stacking and the the best of the configurations that include Voting, for each dataset in which at least one of these Multi-view stacking configurations presented a significant difference with respect to Aggregation (see Table 1). We mark with a check mark if the best of the configurations that include Multi-view stacking presented a significant difference compared to the best of the configurations that include Voting, and if this significant difference was in favor of the best of the configurations that include Multi-view stacking. In the same way, we mark with an *X* mark if the best of the configurations that include Multi-view stacking presented a significant difference compared to the best of the configurations that include Voting, and if this significant difference favored the latter.

In Table 4, we can see that there are significant differences in favor of the best of the configurations that includes Multi-view stacking for most datasets. We also note that there are not significant differences in favor of the best of the configurations that includes Voting. These observations suggest that the Multi-view stacking ensemble could be better or equal than the Voting ensemble.

In Table 5, we can see some important data from the Fingerprint dataset, step 1 of Section 4.4, such as its dimensions and the distribution of its classes. The dimensions of this dataset are 65 rows corresponding to 65 datasets (see Section 4.1) and 210 columns corresponding to the features (defined in step 1 of Section 4.4). Regarding the distribution of classes, we can observe that the class with the greatest number of instances is Aggregation (with 30), followed of Multiviewstacking (with 23), MultiviewstackingNotShuffle (with 11) and Adaboost (with 1). This last observation shows an imbalance of classes.

The result of balancing the Fingerprint dataset are shown in Table 6, step 2 of Section 4.4. In Table 6, we can see that the Fingerprint dataset has the same number of instances (30) per class and, therefore, the number of rows (samples) is now 210.

The results of the recognizing of the best configuration of the fusion strategies are presented in Table 7 and Table 8, corresponding to step 3 of the method described in Section 4.4. In Table 7, we can observe that of 30 Aggregations, RFC predicted that three were MultiviewStackingNotshuffle, two were MultiviewStacking and one was Adaboost. In addition, of 30 MultiviewStakings, RFC predicted that four were MultiviewStackingNotShuffle and two were Aggregation. In addition, out of 30 MultivewStackingNotShuffle, RFC predicted that one was Aggregation. These observations suggest that RFC can predict well MultiviewStackingNot Shuffle and reasonably well Aggregation and MultiviewStacking.

In Table 8, we can see that RFC achieves a precision of 0.90, a recall of 0.89 and an f1-score of 0.89. These observations confirm that RFC can pretty well predict MultiviweStackingNot Shulffle and Aggregation and MultiviewStacking reasonably well.

With an average precision of 90%, we substantiate our claim of a good prediction of the best information integration method, among the four candidates that were among the best in our comparative analysis of the integration method. This is in our view a valuable and original research result.

## 5. Conclusions

In this research, we have proposed to combine information coming from different sensors using an optimal combination method depending on the characteristics of the data to be processed. Our experiments show us that there is no such thing as the “best” sensor combination method for any given sensors and situations, and rather we propose a machine-learning method to find the optimal fusion method for a specific set of data, after an analysis of meta-information about the data that we call its “signature”.

Therefore, we trained a classifier with the signatures information in order to predict, given a dataset not used for the training, which one is the best of the eight strategies considered in this paper (Aggregation, Multi-view stacking (with three configurations), voting (with three arrangements) and Adaboost).

Thus, the contribution of this work is twofold: one is to determine, among a set of information combination methods, which ones in some situations are better than the others (in fact, we reduced the original set of eight combination methods to four), and then we trained a machine-learning classifier to recognize, on the basis of the statistical signatures of our 65 datasets, which one of these four was actually the best one, with an average precision of 90%.

Finally, we have also foreseen a practical architecture for implementing our method with commodity sensors and processors, in such a way that the heaviest processing is carried out offline, and the actual data sensing is done online in a computationally cheaper way.

We believe that the method presented here is the first of its kind, since, so far, we have not found articles that perform a meta-study similar to our work.

We want to make clear that the purpose of this work is not to propose a new human activity recognition method per se (see Section 2.1), but to automate the process of information integration (see Section 2.3), which could or not be associated to HAR, though in this particular work it is the case. In the type of study that we are doing in this paper, what is of paramount importance is the quantity of datasets, so that we can achieve results of statistical significance. This justifies the use of sensors for which there are many datasets publicly available, as is the case for accelerometers and gyroscopes, but not for vision sensors.

Of course, the intention of our automated integration method predictor is not to replace the careful examination of alternatives in a particular application, but only to suggest the most likely effective information integration method, just as automated medical diagnostic systems should not replace a thorough analysis and human doctor judgement. In some cases, though, an automated method for choosing the integration method could be essential, like for instance when the data arriving changes over time, making the manually-chosen integration method less adequate than other ones.

The practical use of the method proposed in this paper assumes the existence of an available collection of datasets publicly available, against which a new dataset will be matched for getting an integration method recommendation. We believe this is a reasonable assumption, given that the reuse of public datasets is more and more done and even encouraged, as it allows to make direct comparison of methods and fosters experimental reproducibility. The more integration methods and more data sets are included in the meta-data set, the greater the probability that a particular data set will be closer to any of the data sets included in the meta-data set.

Regarding the comparison of this paper with other works, we notice that there are many papers for comparing the different activity recognition methods, in different settings and purposes [108,109], but we did not find a single meta-study for which each dataset is a single data point (in our case, each dataset becomes a row with statistical features called “fingerprint”), and then the collection of data points is used for training a classifier, as is the case of our work. Thus, we truly think our study is of a different nature than previous ones.

### Future Work

In the future, we propose to expand our Fingerprint dataset by including more fusion strategies and datasets to recognize a broader set of such strategies and make public this Fingerprint dataset. We also intend to include other activity recognition methods, such as the Markov-based approaches [110], and also include non-activity recognition datasets so that the scope of the work is more general. Another interesting research direction is to consider other aspects beyond accuracy, such as resilience to failures, because, for instance, if an increasing number of failures is detected, the integration method might need to be changed.

We also want to implement and test our virtual sensor architecture with actual electronics, and not only computer simulations.

## Figures and Tables

**Figure 1 sensors-19-02017-f001:**
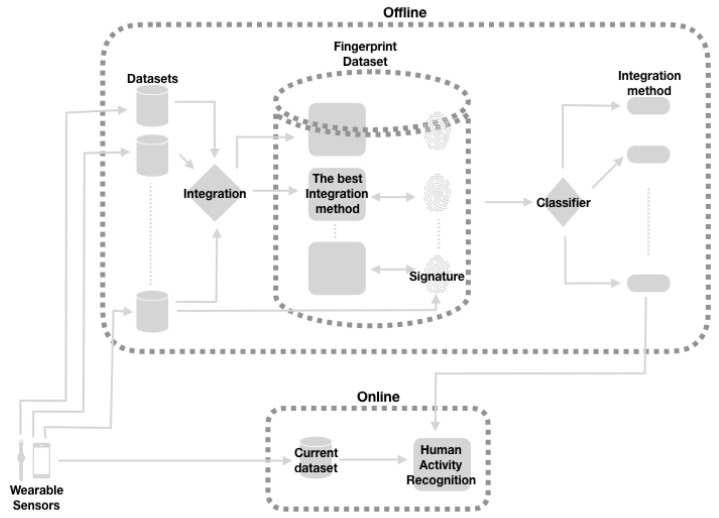
Architecture of a virtual sensor.

**Figure 2 sensors-19-02017-f002:**
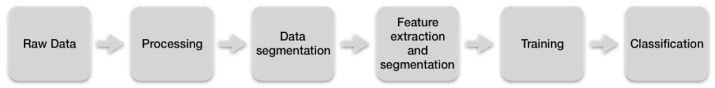
Activity recognition workflow (we adapted it from Bulling et al. [49]).

**Figure 3 sensors-19-02017-f003:**
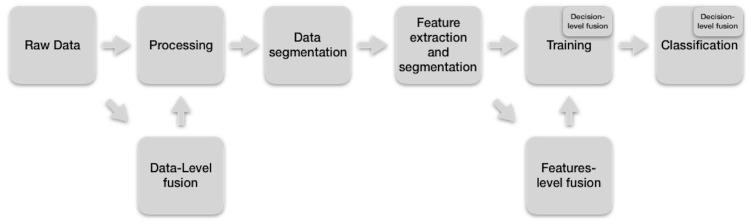
Extended activity recognition workflow.

**Figure 4 sensors-19-02017-f004:**
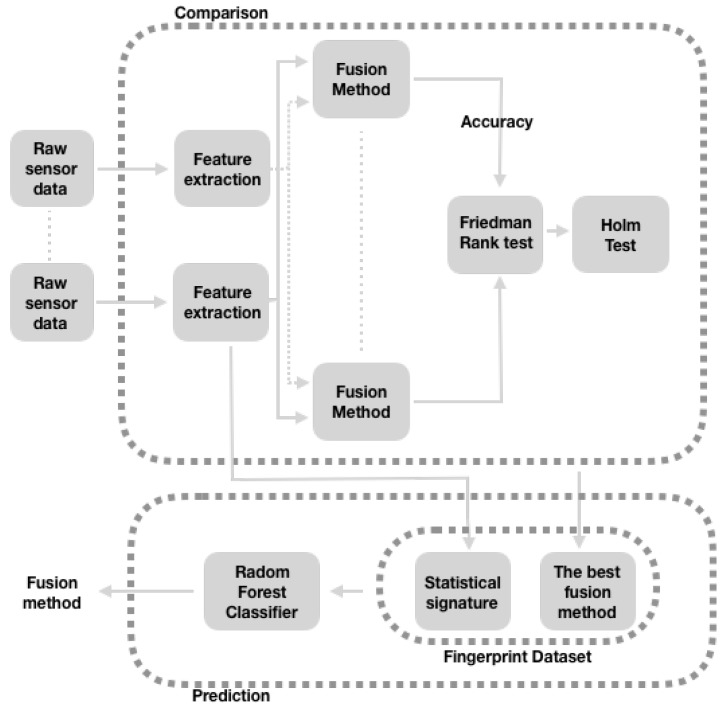
Overview of the optimal fusion method prediction approach.

**Table 1 sensors-19-02017-t001:** Relationship between the configurations of the fusion methods and the datasets, with respect to significant differences found when these configurations mainly used Random Forest as a classifier.

	Configuration	Voting (Shuffled Features)	Voting	Voting All Features CART-LR-RFC	Multi-View Stacking (Shuffle)	Multi-View Stacking	Multi-View Stacking All Features CART-LR-RFC	Adaboost
Dataset								
DailyRlAccRaGy		✘	✘		✔	✔		
PAMAP2		✘	✘	✘				
OpportunityLlAccGy			✘		✘	✘	✘	
DailyLaAccRaGy		✘	✘	✘	✔			
OpportunityLuAccLlGy		✘	✘		✘	✘	✘	
PAMAP2HaAccAnGy		✘	✘	✘				
Opportunity			✘	✘			✘	
DailyLaAccRlGy		✘	✘	✘	✔	✔		
DailyLlAccLaGy		✘	✘	✘	✔	✔		
PAMAP2AnAccChGy		✘	✘	✘				
DailyRaAccLaGy		✘	✘	✘	✔			
OpportunityLlAccLuGy			✘		✘	✘	✘	
OpportunityLlAccRuGy			✘	✘		✘	✘	
DailyLlAccRaGy		✘	✘	✘	✔	✔		
OpportunityRuAccLuGy			✘	✘	✔		✘	
DailyRlAccToGy		✘	✘	✘	✔	✔		
OpportunityRlAccRuGy		✘	✘	✘		✘	✘	
DailyLlAccRlGy		✘	✘	✘	✔			
OpportunityLlAccRlGy			✘	✘		✘	✘	
OpportunityLuAccRuGy		✘	✘	✘			✘	
DailyRaAccRlGy		✘	✘	✘	✔	✔		
DailyLlAccToGy		✘	✘	✘	✔	✔		
DailyRlAccLlGy		✘	✘	✘		✔		
DailySportleftarmAccGy		✘	✘	✘	✔	✔		
DailyRaAccLlGy		✘	✘	✘	✔	✔		
HAPT		✘	✘	✘	✔			
DailySportLeftLegAccGy		✘	✘	✘	✔			
DailyRaAccToGy		✘	✘	✘	✔	✔		
MHealthLaAccRaGy		✔			✔			
OpportunityLuAccRlGy		✘	✘	✘		✘	✘	
DailySport		✘	✘	✘	✔			
OpportunityBaAccLuGy			✘		✔			✔
OpportunityRuAccLlGy			✘	✘			✘	
MHealthRaAccLaGy		✔	✘	✘	✔			
DailyLaAccToGy		✘	✘	✘	✔			
OpportunityLlAccBaGy			✘			✘	✘	
DailySportRightLegAccGy		✘	✘	✘				
MHealth			✘	✘	✔			
DailyLaAccLlGy		✘	✘	✘	✔	✔		
DailyToAccRaGy		✘	✘	✘	✔	✔		
OpportunityBaAccLlGy		✘	✘	✘		✘	✘	
OpportunityRuAccBaGy		✘	✘	✘				
OpportunityBaAccRlGy		✘	✘	✘	✔			
PAMAP2ChAccHaGy		✘	✘	✘		✘		
OpportunityRuAccRlGy		✘	✘	✘	✔		✘	
OpportunityBaAccRuGy		✘	✘		✔			
PAMAP2ChAccAnGy		✘	✘	✘		✘	✘	
PAMAP2AnAccHaGy		✘	✘	✘		✔		
OpportunityRuAccGy		✘	✘	✘			✘	
OpportunityRlAccLlGy			✘					
OpportunityBaAccGy		✘	✘	✘		✘		
PAMAP2ChAccGy		✘	✘	✘		✘		
DailyToAccLlGy		✘	✘	✘	✔			
DailyToAccLaGy		✘	✘	✘	✔			
OpportunityRlAccBaGy		✘	✘	✘		✘		
OpportunityLuAccBaGy		✘	✘	✘		✘	✘	
MHealthLaAccGy		✔		✘	✔		✘	
OpportunityRlAccLuGy		✘	✘	✘			✘	
PAMAP2HaAccChGy		✘	✘	✘	✔	✔		
PAMAP2AnAccGy		✘	✘	✘		✘	✘	✘
OpportunityLuAccGy		✘	✘	✘	✘	✘	✘	
UDT		✘	✘	✘		✘	✘	
DailyToAccRlGy		✘	✘	✘	✔			
DailyRlAccLaGy		✘	✘	✘		✔		
DailySportTorsoAccGy		✘	✘	✘	✔			

**Table 2 sensors-19-02017-t002:** Relationship between the configurations of the fusion methods and the datasets, with respect to the largest significant differences found when these configurations used Random Forest as a classifier.

	Configuration	Voting (Shuffled Features)	Voting	Voting All Features CART-LR-RFC	Multi-View Stacking (Shuffle)	Multi-View Stacking	Multi-View Stacking All Features CART-LR-RFC	Adaboost
Dataset								
DailyRlAccRaGy						✔		
PAMAP2								
OpportunityLlAccGy								
DailyLaAccRaGy					✔			
OpportunityLuAccLlGy								
PAMAP2HaAccAnGy								
Opportunity								
DailyLaAccRlGy						✔		
DailyLlAccLaGy						✔		
PAMAP2AnAccChGy								
DailyRaAccLaGy					✔			
OpportunityLlAccLuGy								
OpportunityLlAccRuGy								
DailyLlAccRaGy						✔		
OpportunityRuAccLuGy					✔			
DailyRlAccToGy						✔		
OpportunityRlAccRuGy								
DailyLlAccRlGy					✔			
OpportunityLlAccRlGy								
OpportunityLuAccRuGy								
DailyRaAccRlGy						✔		
DailyLlAccToGy					✔			
DailyRlAccLlGy						✔		
DailySportleftarmAccGy								
DailyRaAccLlGy					✔			
HAPT					✔			
DailySportLeftLegAccGy					✔			
DailyRaAccToGy					✔			
MHealthLaAccRaGy					✔			
OpportunityLuAccRlGy								
DailySport					✔			
OpportunityBaAccLuGy								✔
OpportunityRuAccLlGy								
MHealthRaAccLaGy					✔			
DailyLaAccToGy					✔			
OpportunityLlAccBaGy								
DailySportRightLegAccGy								
MHealth					✔			
DailyLaAccLlGy						✔		
DailyToAccRaGy					✔			
OpportunityBaAccLlGy								
OpportunityRuAccBaGy								
OpportunityBaAccRlGy					✔			
PAMAP2ChAccHaGy								
OpportunityRuAccRlGy					✔			
OpportunityBaAccRuGy					✔			
PAMAP2ChAccAnGy								
PAMAP2AnAccHaGy						✔		
OpportunityRuAccGy								
OpportunityRlAccLlGy								
OpportunityBaAccGy								
PAMAP2ChAccGy								
DailyToAccLlGy					✔			
DailyToAccLaGy					✔			
OpportunityRlAccBaGy								
OpportunityLuAccBaGy								
MHealthLaAccGy					✔			
OpportunityRlAccLuGy								
PAMAP2HaAccChGy						✔		
PAMAP2AnAccGy								
OpportunityLuAccGy								
UDT								
DailyToAccRlGy					✔			
DailyRlAccLaGy						✔		
DailySportTorsoAccGy					✔			

**Table 3 sensors-19-02017-t003:** Relationship between the Multi-view stacking configuration that shuffles features and the datasets, with respect to the significant differences found when this configuration is compared to the Multi-view stacking configuration that does not shuffle features.

	Configuration	Multi-View Stacking (Shuffle)
Dataset		
DailyRlAccRaGy		
DailyLaAccRaGy		
DailyLaAccRlGy		
DailyLlAccLaGy		
DailyRaAccLaGy		
DailyLlAccRaGy		
OpportunityRuAccLuGy		
DailyRlAccToGy		
DailyLlAccRlGy		
DailyRaAccRlGy		
DailyLlAccToGy		
DailyRlAccLlGy		
DailyRaAccLlGy		
HAPT		
DailySportLeftLegAccGy		✔
DailyRaAccToGy		
MHealthLaAccRaGy		✔
DailySport		✔
MHealthRaAccLaGy		✔
DailyLaAccToGy		
MHealth		
DailyLaAccLlGy		
DailyToAccRaGy		
OpportunityBaAccRlGy		✔
OpportunityRuAccRlGy		
OpportunityBaAccRuGy		✔
PAMAP2AnAccHaGy		
DailyToAccLlGy		✔
DailyToAccLaGy		
MHealthLaAccGy		✔
PAMAP2HaAccChGy		
DailyToAccRlGy		✔
DailyRlAccLaGy		
DailySportTorsoAccGy		✔

**Table 4 sensors-19-02017-t004:** Summary of the significant differences between the best of the configurations that include Multi-view stacking and the best of the configurations that include Voting.

	Configuration	The Best Multi-View Stacking	The Best Voting	Comparison
Dataset				
DailyRlAccRaGy		MultiViewStackingNotShuffle	VoteAllFeatures	✔
DailyLaAccRaGy		MultiViewStacking	Vote	✔
DailyLaAccRlGy		MultiViewStackingNotShuffle	Vote	✔
DailyLlAccLaGy		MultiViewStackingNotShuffle	Vote	✔
DailyRaAccLaGy		MultiViewStacking	Vote	✔
DailyLlAccRaGy		MultiViewStackingNotShuffle	VoteAllFeatures	✔
OpportunityRuAccLuGy		MultiViewStacking	Vote	✔
DailyRlAccToGy		MultiViewStackingNotShuffle	Vote	✔
DailyLlAccRlGy		MultiViewStacking	Vote	✔
DailyRaAccRlGy		MultiViewStackingNotShuffle	Vote	✔
DailyLlAccToGy		MultiViewStacking	Vote	✔
DailyRlAccLlGy		MultiViewStackingNotShuffle	VoteAllFeatures	✔
DailyRaAccLlGy		MultiViewStacking	Vote	✔
HAPT		MultiViewStacking	Vote	✔
DailySportLeftLegAccGy		MultiViewStacking	Vote	✔
DailyRaAccToGy		MultiViewStacking	Vote	✔
MHealthLaAccRaGy		MultiViewStacking	Vote	
DailySport		MultiViewStacking	VoteAllFeatures	✔
MHealthRaAccLaGy		MultiViewStacking	Vote	
DailyLaAccToGy		MultiViewStacking	Vote	✔
MHealth		MultiViewStacking	Vote	
DailyLaAccLlGy		MultiViewStackingNotShuffle	Vote	✔
DailyToAccRaGy		MultiViewStacking	Vote	✔
OpportunityBaAccRlGy		MultiViewStacking	Vote	✔
OpportunityRuAccRlGy		MultiViewStacking	Vote	✔
OpportunityBaAccRuGy		MultiViewStacking	VoteAllFeatures	✔
PAMAP2AnAccHaGy		MultiViewStackingNotShuffle	Vote	✔
DailyToAccLlGy		MultiViewStacking	Vote	✔
DailyToAccLaGy		MultiViewStacking	Vote	✔
MHealthLaAccGy		MultiViewStacking	Vote	
PAMAP2HaAccChGy		MultiViewStackingNotShuffle	Vote	✔
DailyToAccRlGy		MultiViewStacking	Vote	✔
DailyRlAccLaGy		MultiViewStackingNotShuffle	VoteAllFeatures	✔
DailySportTorsoAccGy		MultiViewStacking	Vote	✔

**Table 5 sensors-19-02017-t005:** Important data on the Fingerprint dataset.

Dataset	Dimensions	Class Distribution			
		Aggregation	Multi-View	Multi-View	Adaboost
			Stacking	Stacking NotShuffle	
Fingerprint	(65, 210)	30	23	11	1

**Table 6 sensors-19-02017-t006:** Balanced Fingerprint dataset.

Dataset	Dimensions	Class Distribution			
		Aggregation	Multi-View	Multi-View	Adaboost
			Stacking	Stacking NotShuffle	
Fingerprint	(120, 210)	30	30	30	30

**Table 7 sensors-19-02017-t007:** Confusion matrix of Multi-view stacking with RFC on a Fingerprint Dataset.

Label	Adaboost	Aggregation	Multi-View Stacking	Multi-View Stacking NotShuffle
Adaboost	30	0	0	0
Aggregation	1	24	2	3
MultiViewStacking	0	2	24	4
MultiViewStackingNotShuffle	0	1	0	29

**Table 8 sensors-19-02017-t008:** Metrics of Multi-view stacking with RFC on the Fingerprint Dataset.

Label	Precision	Recall	f1-Score	Support
Adaboost	0.97	1.00	0.98	30
Aggregation	0.89	0.80	0.84	30
MultiViewStacking	0.92	0.80	0.86	30
MultiViewStackingNotShuffle	0.81	0.97	0.88	30
avg/total	0.90	0.89	0.89	120

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
