# Peer review of "Virtual Sensors for Optimal Integration of Human Activity Data"

_sensors, 2019, doi:10.3390/s19092017_

Round 1

Reviewer 1 Report

The authors present a method to suitably define a combination of features extracted form gyroscopes and accelerometers for human activity recognition. The authors select features and sensor fusion techniques that are commonly successful for human activity recognition. Moreover, they propose shuffling of features to test whether this action has an effect. They then train a classifier of the best data processing technique based on 65 datasets that they obtained from 6 well-known human activity recognition datasets.

The paper is organized sufficiently well, but, in my opinion, there are major issues that limit the value and the potential impact of the work.

1) The authors do not explain how to transfer results to other HAR cases: how could a new HAR application be said to be “close” to the one represented by the datasets that you took into account?

2) According to the title, the classifier should work with any virtual sensor based on any combination of physical sensors, any set of features and any application (not only HAR)

3) I am concerned with the fact that if the application changes the whole procedure of finding virtual sensors should be done from scratch, is it worth it for a moderate improvement? This makes me think that the value of this method is limited

4) What do the authors mean by high-precision? What is the practical effect of moving from 95% to 96% accuracy?

5) How the method is sensitive to mistakes. Using machine-learning methods hinder the possibility to investigate the amount of error due to selecting one method in place of a better one.

In addition to these major issues, I have other concerns:

6) The title does not reflect the scope of the paper. It should be focused on gyro+accelerometers and HAR application case.

7) How relevant is selecting gyroscopes and accelerometers, rather than vision sensors, in HAR

8) Why other activity recognition approaches, such as the ones based on Markov Models (e.g. partially observed Markov Decision Process)

9) Minor issues: page 5, LSTM stands for long-short-term memory. In the results of the Friedmann tests, please report p-values smaller than 10e-3 as p<0.001. There’s no need to have more than 3 figures after the comma, as alpha values are expressed as 0.05, 0.01, or 0.001  

Author Response

Reviewer # 1

The authors present a method to suitably define a combination of features extracted form gyroscopes and accelerometers for human activity recognition. The authors select features and sensor fusion techniques that are commonly successful for human activity recognition. Moreover, they propose shuffling of features to test whether this action has an effect. They then train a classifier of the best data processing technique based on 65 datasets that they obtained from 6 well­known human activity recognition datasets.

The paper is organized sufficiently well, but, in my opinion, there are major issues that limit the value and the potential impact of the work.

1) The authors do not explain how to transfer results to other HAR cases: how could a new HAR application be said to be “close” to the one represented by the datasets that you took into account?

Answer to this observation by authors:

We added a related explanation to the paper (subsection 4.1, page 23, lines 583-605). The idea is that as more integration methods and more data sets are included in the meta-data set (we have right now 65 datasets), the greater the probability that a particular data set will be closer to any of the data sets included in the meta-data set.

2) According to the title, the classifier should work with any virtual sensor based on any combination of physical sensors, any set of features and any application (not only HAR)

Answer to this observation by authors:

We are changing the title in order to reflect the fact that right now the scope of our datasets is HAR; the updated title is: “Virtual Sensors for Optimal Integration of Human Activity Data collected by Wearable Sensors”. We want to notice that the methods we are developing are in principle general, but given that they have been validated just in HAR, we agree to reduce the scope, as reflected in the new title.

3) I am concerned with the fact that if the application changes the whole procedure of finding virtual sensors should be done from scratch, is it worth it for a moderate improvement? This makes me think that the value of this method is limited

Answer to this observation by authors:

This practical concern is indeed valid, but we are assuming that a public collection of datasets could be made available, so that each individual author is not starting “from scratch”, but reusing existing resources. In many Pattern Recognition areas the reuse of existing databases is not only allowed but also encouraged, because it allows to make useful performance comparisons of different methods over the same data. This is discussed in section 4.1, page 23, lines 599-605.

4) What do the authors mean by high ­precision? What is the practical effect of moving from 95% to 96% accuracy?

Answer to this observation by authors:

We are replacing the mentions of the term “high precision” by a specific figure to avoid vague language.

Now, about the practical effect of improving from 95% to 96%, we have two comments: first, in some cases (now reported in the paper, see lines 74-76) the improvement is of as much of 4 percent points (88% for voting to 92% for stacking). Second, the importance of a small improvement depends on the application area, which is well outside of the scope of this paper. In medical critical applications, or in tasks repeated millions of times, a small improvement could mean a substantial benefit.

5) How the method is sensitive to mistakes. Using machine ­learning methods hinder the possibility to investigate the amount of error due to selecting one method in place of a better one.

Answer to this observation by authors:

We are not proposing the replace the careful examination and comparison of different options, but to have a suggestion of which methods are likely to perform better according to the ML-trained predictor. The fact that our prediction has a precision above 90% means that it’s reasonable to take this prediction into account as a starting point for such a method comparison. We have to keep in mind that the prediction made by the system has a statistical basis, and it is in principle possible that someone works with a highly atypical dataset, not similar to any dataset included for training the classifier, and then such a case will fall in the <10% margin or error of our proposed method. This is commented in the Discussion and related work subsection (page 23, lines 593-605).

6) The title does not reflect the scope of the paper. It should be focused on gyro+accelerometers and HAR application case.

Answer to this observation by authors:

The title has been modified, as commented above.

7) How relevant is selecting gyroscopes and accelerometers, rather than vision sensors, in HAR

Answer to this observation by authors:

Please keep in mind that our method relies on the existence of a large number of datasets, which is the case for gyroscopes and accelerometers, and not, for instance, for vision sensors. Without a good number of datasets we can’t even check for statistical significance, which we do in this paper. According to Demsar (“Statistical comparisons of classifiers over multiple data sets”, 2006), when comparing multiple classifiers, it is essential to perform tests on as many datasets as possible. Although we do not directly compare the classifiers, we do compare them indirectly, since the integration methods use classifiers in their internal processes, so, for practical purposes, the results of Demsar are valid in our comparisons (see section 4.1, page 22-23, lines 583-592).

8) Why other activity recognition approaches, such as the ones based on Markov Models (e.g. partially observed Markov Decision Process)

Answer to this observation by authors:

Take into account that the focus of this work is not the whole area of activity recognition methods, but just the information integration methods. Now, of course the use of approaches such as POMDP could be integrated into our method as building blocks, just as the results of classifiers are used in the current paper; this is now mentioned in the paper in the Future work section (page 23, lines 634-637).

9) Minor issues: page 5, LSTM stands for long ­short­ term memory. In the results of the Friedmann tests, please report p­values smaller than 10e­3 as p<0.001. There’s no need to have more than 3 figures after the comma, as alpha values are expressed as 0.05, 0.01, or 0.001

Answer to this observation by authors:

A thorough terminology, grammatical and spelling revision has been made, in particular we corrected LSTM in page 5, line 161.

We agree that alpha values are not reported in the literature with lots of decimals, and we have changed in the paper the way they are reported, just mentioning in the text when they fall below the 0.05 margin (in particular on pages 12-13, lines 455-465), though we report some decimals in the tables.

Reviewer 2 Report

The paper shows how  to choose the best fusion approach for action recognition.

1. Authors use the term activity however, based on the datsets they used, I prefer to use the term action.

2. Algorithm 1 in page 10 is not really an algorithm. Authors may present their platform in an appropriate way

3. Authors should add a discussion section which compares their results with other works from the-state-of-athe-art.

4. Some minor typos and english corrections should be done

Author Response

Reviewer # 2

The paper shows how to choose the best fusion approach for action recognition.

1. Authors use the term activity however, based on the datasets they used, I prefer to use the term action.

Answer to this observation by authors:

A clarification about the term “activity” is now included as a footnote in page 2.

2. Algorithm 1 in page 10 is not really an algorithm. Authors may present their platform in an appropriate way

Answer to this observation by authors:

Of course you are right. We have removed the wrongly called “algorithm” and added some explanations related to implementation (pages 9-10, lines 353-364).

3. Authors should add a discussion section which compares their results with other works from the ­state-of-­the-­art.

Answer to this observation by authors:

We have added a new subsection, 4.1, to deal with discussion of results and related work comparison, but we claim that the current paper is the first one that presents a meta-study for which each dataset leads to a row in a dataset of statistical signatures, from which our predictor is trained. No other such study exists now, to the best of our knowledge.

4. Some minor typos and english corrections should be done

Answer to this observation by authors:

A thorough terminology, grammatical and spelling revision has been made.

Round 2

Reviewer 1 Report

I see the effort of the authors to explain their approach. However, I still have the same doubts I had when reading the (recent) previous version. A practical example of how using the aggregation method on a new case could either shorten, speed-up, improve (even better exploiting existing datasets) the whole process of HAR (I see that HAR is only a use case, but this is the only one the authors proposed so far) would make me rethink about the method. In my opinion, the paper is not publishable in the present state.

Author Response

Answer to reviewers observations 

Reviewer #1

“I see the effort of the authors to explain their approach. However, I. A practical example of how using the aggregation method on a new case could either shorten, speed-up, improve (even better exploiting existing datasets) the whole process of HAR (I see that HAR is only a use case, but this is the only one the authors proposed so far) would make me rethink about the method. In my opinion, the paper is not publishable in the present state.”

Answer to this observation by authors: 

This new review is so short and unspecific that it appears to have been written in a hurry; its main flaw is that it fails to give us a specific point to correct or improve. Further, it’s vague when saying that “still have the same doubts I had when reading the (recent) previous version,” but not writing down which these doubts are.

After that, the same reviewer says “However, I. A practical example of how using the aggregation method on a new case could either shorten, speed-up, improve (even better exploiting existing datasets) the whole process of HAR (I see that HAR is only a use case, but this is the only one the authors proposed so far) would make me rethink about the method.” This remark is puzzling because, beyond the poor phrase construction, it implies that we are proposing to improve the aggregation method when used for HAR, which shows a misunderstanding of the essence of our method because we use the aggregation method just as the baseline lowest reference, not as an improvement over anything. We believe that the comment from the reviewer seems to refer to a different idea than our actual proposal.

The final remark, “In my opinion, the paper is not publishable in the present state,” could be perfectly valid as an assessment of a paper, except that is not substantiated with objective evidence.
